# Are Women with Normal-Weight Obesity at Higher Risk for Cardiometabolic Disorders?

**DOI:** 10.3390/biomedicines11020341

**Published:** 2023-01-25

**Authors:** Damoon Ashtary-Larky, Sara Niknam, Meysam Alipour, Reza Bagheri, Omid Asbaghi, Mehrnaz Mohammadian, Salvador J. Jaime, Julien S. Baker, Alexei Wong, Katsuhiko Suzuki, Reza Afrisham

**Affiliations:** 1Nutrition and Metabolic Diseases Research Center, Ahvaz Jundishapur University of Medical Sciences, Ahvaz 61357-15794, Iran; 2Department of Clinical Biochemistry, School of Medicine, Isfahan University of Medical Sciences, Isfahan 81746-73461, Iran; 3Department of Nutrition, Shoushtar Faculty of Medical Sciences, Shoushtar 38196-93345, Iran; 4Department of Exercise Physiology, University of Isfahan, Isfahan 81746-73441, Iran; 5Cancer Research Center, Shahid Beheshti University of Medical Sciences, Tehran 14167-53955, Iran; 6Department of Exercise Physiology, Islamic Azad University of Ahvaz, Ahvaz 61349-37333, Iran; 7University of Wisconsin-La Crosse, La Crosse, WI 54601, USA; 8Centre for Health and Exercise Science Research, Department of Sport and Physical Education, Hong Kong Baptist University, Kowloon Tong 999077, Hong Kong; 9Department of Health and Human Performance, Marymount University, Arlington, VA 22207, USA; 10Faculty of Sport Sciences, Waseda University, 2-579-15 Mikajima, Tokorozawa 359-1192, Japan; 11Department of Clinical Laboratory Sciences, Faculty of Allied Medicine, Tehran University of Medical Sciences, Tehran 14177-44361, Iran

**Keywords:** body mass index, normal-weight obesity, cardiometabolic disease, cardiovascular disease, obesity, body fat percentage

## Abstract

Objectives: This study aimed to evaluate the cardiometabolic abnormalities in women with normal-weight obesity (NWO) in comparison with lean, overweight, and obese women. Methods: This cross-sectional study evaluated the assessment of cardiometabolic abnormalities of women with NWO compared to lean, overweight, and obese women. NWO was defined as a BMI < 25 kg.m^−2^ and a BFP higher than 30%. Anthropometric variables, cardiometabolic abnormality markers (fasting blood glucose (FBG), blood pressure (BP), lipid profile, insulin resistance, and high-sensitivity C-reactive protein (hs-CRP)), and liver enzymes were also examined. Results: Significant differences were observed in HDL concentrations between NWO, lean, and obese participants (*p* < 0.05). There were no significant differences in FBG, insulin resistance, liver enzymes, or cholesterol between groups (*p* > 0.05). The prevalence of the abnormal metabolic phenotype was higher in NWO compared to the lean group (4.0% and 24.1%, respectively; *p* < 0.05). Women with type 2 and 3 obesity had abnormal metabolic profiles (60.9% and 73.9%, respectively) compared to NWO participants (*p* < 0.01). The NWO group had a significantly higher incidence of cardiometabolic abnormalities compared to the lean participants (*p* < 0.05), while the type 2 and 3 obese individuals had significantly higher incidences compared to the NWO group (*p* < 0.001 and *p* < 0.001, respectively). Conclusions: Individuals with NWO had a significantly higher incidence of cardiometabolic abnormalities when compared to lean participants. These abnormalities strongly relate to BFP and waist circumferences.

## 1. Introduction

In recent decades, the significant increase in the global prevalence of obesity has been strongly associated with an increased risk of developing a wide range of chronic diseases such as diabetes, cardiovascular disease, dyslipidemia, inflammatory disease, depression, and cancer [1,2,3,4]. The World Health Organization (WHO) defines obesity as an excessive accumulation of fat to a level that may compromise health [5]. Although the body mass index (BMI) is a simple index of weight-to-height ratio that is commonly used to classify overweight and obese phenotypes, evidence suggests that the complexity of obesity cannot be characterized by BMI in isolation [6,7]. Therefore, to elucidate the relationship between obesity and disease development, it is imperative to measure fat mass (FM). It has been observed that lean and obese participants have different metabolic characteristics [8]. Moreover, body fat percentage (BFP) and FM distribution may be different in participants with the same BMI.

This inherent limitation of BMI in the accurate diagnosis of obesity has been observed to a greater extent among Asians and the elderly, who commonly have lower lean body mass (LBM) [9,10]. Consequently, the concept of normal-weight obesity (NWO) has been proposed. The term “NWO” describes individuals with normal body weight and BMI (<25 kg.m^−2^) who have increased BFP (>30% in females) [11,12,13]. Previous studies involving middle-aged and older (35–75 years) adults as well as wide-age-range (16–75 years of age) cohorts concluded that, despite their normal weights, NWO participants have similar FM distributions and risks of developing cardiovascular diseases, type 2 diabetes, and metabolic syndrome as overweight and obese women [14,15,16]. However, the risk for cardiometabolic abnormalities in young (<35 years of age) women with NWO has not been established. The prevalence of NWO is significantly greater in women, and higher risks seem to be caused by increased FM [15,17,18,19,20,21,22]. Specific screening and treatment for NWO in young women may be clinically relevant to implement cardiometabolic prevention and treatment at an early age. Because of the positive effects of exercise training and proper nutrition on cardiometabolic biomarkers as well as increasing LBM and decreasing BFP [23,24,25,26], it seems that lifestyle intervention by adding exercise and appropriate nutrition can be a therapeutic approach for NWO. 

The purpose of this study was to investigate the anthropometric and cardiometabolic profiles of young women with NWO. In addition, we assessed the relationship between BFP and cardiometabolic abnormalities in young women with NWO in comparison with lean, overweight, and obese young women. We hypothesized that young women with NWO would have high incidences of cardiometabolic abnormalities and therefore elevated disease risks when compared to lean participants.

## 2. Methods

### 2.1. Ethical Approval and Consent to Participate

The study protocols were carried out in compliance with the Declaration of Helsinki and were approved by the Ethics Committee of Ahvaz Jundishapur University of Medical Sciences (ID number: IR.AJUMS.REC.1394.489, approval date: 12 May 2015). Written informed consent was obtained from all participants prior to the study.

### 2.2. Participants

We studied 154 young sedentary women, aged 20–35 years, in a cross-sectional study. Individuals were categorized into six groups, depending on their respective BMI scores and BFP:24 lean women (18.5 ≤ BMI < 25 and BFP < 30%);29 NWO women (18.5 ≤ BMI < 25 and BFP > 30%);28 overweight women (25 ≤ BMI < 30);27 type 1 obese women (30 ≤ BMI < 35);23 type 2 obese women (35 ≤ BMI < 40);23 type 3 obese women (BMI ≥ 40).

Women with NWO were distinguished from lean women based on fat mass distribution, which was determined using the bioelectrical impedance method, namely the BFP classification criterion.

Young sedentary women who were referred to a nutrition clinic in Ahvaz, Iran, were recruited as potential participants and screened based on inclusion/exclusion criteria. The inclusion criteria included having regular 28 d menstrual cycles, a lack of physical activity, no smoking, no alcohol consumption, no usage of any supplements, and a lack of weight change in the previous 6 months. The exclusion criteria included pregnancy; breastfeeding; the use of any drugs; eating disorders; diabetes; cardiovascular disease; kidney problems; thyroid, digestive, and respiratory diseases; and cancer. Participants consuming more than 300 mg of caffeine daily (described as caffeine users) [27] were excluded from the study.

### 2.3. Anthropometric and Blood Pressure Measurements

Measurements of body weight and body composition were performed using the direct multifrequency bioelectrical impedance method (Inbody 270, Biospace, Seoul, Korea). Waist circumference (WC) was measured at the site of noticeable waist narrowing, located approximately halfway between the costal border and the iliac crest. Hip circumference (HC) was obtained approximately at the level of the pubis symphysis, in the region of the greatest posterior protuberance [17]. Participants were asked to fast for 12 h overnight [28], with at least 8 h of sleep, and avoid any vigorous physical activity for the previous 36 h before the anthropometric measurements [29]. The participants were also instructed to avoid exercising and consuming alcohol for 48 h before testing [30]. Blood pressure (BP) was measured using an automatic monitor (BM65, Beurer, Swabia, Germany) after participants were rested, relaxed, and seated for at least 10 min. The measurements were recorded in triplicate, and the mean value was calculated for each participant.

### 2.4. Blood Sampling

Blood samples were collected in the morning following a 12 h fasting period. Blood sampling collection was performed between days 8 and 12 of the preovulation phase for each participant [11] to minimize the potential effects of endogenous estrogens on metabolic markers [31]. Serum concentrations of glucose, cholesterol, triacylglycerol (TG), high-density lipoprotein (HDL), and low-density lipoprotein (LDL) were measured spectrophotometrically [32] (Pars Azmoon Inc., Tehran, Iran) using an auto-analyzer (Hitachi, Tokyo, Japan). The inter-and intra-assay coefficients of variation (CVs) were less than 2.2% for glucose. The inter- and intra-assay CVs were 2 and 0.5% for HDL and LDL and 1.6 and 0.6% for triglyceride, respectively. Alkaline phosphatase (ALP), aspartate aminotransferase (AST), and alanine aminotransferase (ALT) levels were assayed using the mentioned kits (Pars Azmun Inc., Tehran, Iran). All inter- and intra-assay coefficients of variation were <5%. Fasting serum insulin was assayed using an ELISA kit (DiaPlus inc., North York, Ontario, Canada), with intra- and inter-assay CVs of 4.9% and 8%, respectively. High-sensitivity C-reactive protein (hs-CRP) was measured using ELISA kits (Diagnostic Biochem, Ontorio, Canada), with intra- and inter-assay CVs of 7.7 and 9.7%, respectively. Insulin resistance (homeostatic model assessment of insulin resistance (HOMA-IR)), pancreatic β-cell function (homeostatic model assessment of β-cell function (HOMA-B) percentage), and insulin sensitivity (homeostatic model assessment of insulin sensitivity (HOMA-S) percentage) were estimated as previously described [17,33,34]. All biochemical assays were performed in duplicate, and the means was calculated for each participant.

### 2.5. Definition of Cardiometabolic Risk Factors

Six cardiometabolic abnormalities were considered using the definition recommended by Wildman et al. [35] (Table 1). This gave a more comprehensive representation of an individual’s metabolic health by incorporating components of metabolic syndrome, insulin resistance, and systemic inflammation. Participants were classified as metabolically abnormal if they fulfilled two cardiometabolic abnormalities listed in Table 1. Participants were classified as metabolically healthy if they fulfilled less than two cardiometabolic abnormalities.

### 2.6. Statistical Analysis

An a priori sample size calculation was conducted using the G*Power analysis software [36]. Our rationale for sample size was based on previous studies that evaluated the effects of NWO on metabolic abnormalities [11,13,14,37,38,39]. Based on α = 0.05, effect size = 0.5, and a power (1 − β) of 0.8, the analysis revealed that a total sample size of at least 60 participants (n = 10 per group) was needed to have sufficient power to detect significant changes in the effects of NWO on metabolic abnormalities. All statistical analyses were performed using IBM SPSS Statistics software version 24 (IBM SPSS Statistics, Armonk, NY, USA). The normality of variables was confirmed using Kolmogorov–Smirnov tests. A one-way ANOVA was used to assess differences within groups. Comparisons of variables that were different between the groups were carried out using post hoc Tukey tests. A stepwise linear regression was applied to assess BFP and WC associations with factors of cardiometabolic abnormalities. The alpha level was set at 0.05. The chi-squared test was used to evaluate differences within groups for qualitative variables (cardiometabolic abnormality phenotype). Moreover, Pearson correlations were applied to assess the relationships between WC and BFP and the cardiometabolic markers.

## 3. Results

### 3.1. Physical Characteristics and BP

The anthropometrics, body composition, and BP of all groups are shown in Table 2. As expected, weight, WC, HC, BMI, FM, and BFP were significantly different between all groups (*p* < 0.01). There were significant differences for LBM (*p* < 0.05, *p* < 0.001) in the NWO group compared to the overweight and obese groups, and there were no significant differences between the NWO and lean groups (*p* = 0.638, *p* = 0.380). However, the LBM percentage (LBM%) was significantly different (*p* < 0.001) in the NWO group compared to the lean and obese (types 1, 2, and 3) groups but not the overweight group. No significant differences in heart rate (HR) or diastolic blood pressure (DBP) were evident between the six groups (*p* > 0.05). However, systolic blood pressure (SBP) was significantly higher in participants with type 2 and 3 obesity compared to lean participants (*p* < 0.05).

### 3.2. Cardiometabolic and Liver Enzymes

As shown in Table 3, there were no significant differences in fasting blood glucose (FBG), insulin, homeostatic model assessment of insulin resistance (HOMA-IR), homeostatic model assessment of insulin sensitivity (HOMA-S), or homeostatic model assessment of β-cell function (HOMA-B) between NWO and lean, overweight, or obese participants (*p* > 0.05). However, there were significant differences in FBG (*p* < 0.05), insulin (*p* < 0.05), and HOMA-IR (*p* < 0.01) between lean and type 3 obese participants. There were no significant differences in TG, LDL, LDL/HDL, cholesterol/HDL, or very low density lipoprotein (VLDL) between NWO participants and lean, overweight, or obese participants (*p* > 0.05). HDL was significantly greater in NWO compared to both type 2 (*p* < 0.05) and type 3 (*p* < 0.001) obese participants but lower than in lean participants (*p* < 0.05). Lean participants had significantly lower concentrations of total cholesterol compared to type 3 obese participants (*p* < 0.01). TG concentrations were significantly higher in type 3 obese participants compared to lean (*p* < 0.01), NWO (*p* < 0.01), and overweight (*p* < 0.05) participants.

### 3.3. Correlations between WC and BFP and the Cardiometabolic Markers

The differences in AST, ALT, and ALP levels were not significant (*p* > 0.05). The plasma concentration of hs-CRP was significantly lower in the lean (*p* < 0.001, *p* < 0.001), NWO (*p* < 0.001, *p* < 0.001), overweight (*p* < 0.001, *p* < 0.001), and type 1 obese (*p* < 0.01, *p* < 0.001) groups compared to the type 2 and 3 obese groups. In addition, partial correlations between BFP and WC and the cardiometabolic abnormalities are shown in Table 4. Moreover, according to the results of the Pearson correlation test, significant correlations were found between various cardiometabolic markers and WC and BFP, which are reported in Appendix A.

### 3.4. Between-Group Comparison of the Cardiometabolic Abnormalities

Cardiometabolic abnormalities can be seen in Figure 1. The NWO group had a significantly higher incidence of cardiometabolic abnormalities compared to the lean participants (*p* < 0.05), whereas the type 2 and 3 obese groups were significantly higher compared to the NWO group (*p* < 0.001 and *p* < 0.001, respectively). There were not any significant differences in cardiometabolic abnormalities among NWO women compared to the overweight and type 1 obesity groups.

## 4. Discussion

The main purpose of this study was to identify differences in cardiometabolic abnormalities between NWO, lean, overweight, and obese (types 1–3) young women. Furthermore, we evaluated body composition and liver enzymes in these six groups. Our results show that, compared to lean women, the increased BFP observed among NWO young women was associated with increased levels of cardiometabolic risk, indicating that NWO young women are at an elevated risk of developing cardiometabolic diseases. Compared to NWO participants, type 2 and 3 obese individuals had significantly higher cardiometabolic abnormalities. These abnormalities were strongly related to BFP and WC. Moreover, cardiometabolic abnormalities were statistically the same between women with NWO and overweight and type 1 obesity. No significant differences were observed for liver enzymes, insulin sensitivity, or beta-cell function between lean, NWO, overweight, and obese young women.

Although both lean and NWO participants displayed lean BMI values (18.5–25), our results illustrate a significant difference between the weights of the lean and NWO participants. In addition, most of the lean sedentary women in our study had BMI values lower than 22 (between 18.5 and 22). These findings suggest that sedentary women in the higher range of normal BMI (higher than 22–23) are more likely to be NWO, which is consistent with almost all previous studies [9,11,40,41]. NWO has been characterized as having low lean mass despite having a similar BMI compared to lean participants [41]. There is increasing evidence that in some Asian populations a specific BMI reflects a higher percentage of body fat and less muscle mass than in white or European populations [42]. According to our findings, although the absolute LBM was not statistically different between NWO and lean individuals, the relative LBM (%) was significantly lower in NWO compared to lean participants. A lower relative LBM in NWO women compared with lean women was reported in previous studies [11,14]. It has been shown that low muscle mass is associated with cardiometabolic risk [43,44], regardless of nutritional status [45]. Low muscle mass has been recognized as a disease in the ICD-10 (the International Statistical Classification of Disease and Related Health Problems) [40]. Therefore, the lower LBM in NWO, to some extent, may be the reason for higher cardiometabolic risk compared to lean participants. Due to the higher prevalence of NWO (higher FM and LBM) in the Asian population, according to the WHO, available data do not necessarily indicate a clear BMI cut-off point for all Asians for overweight or obesity, and the classic BMI classification is not appropriate for some Asian participants [46]. Based on the data available in Asia, a WHO expert consultation concluded that, for many Asian populations, additional trigger points for public health action were identified, with 23 kg·m^−2^ or higher representing increased risk and 27.5 kg·m^−2^ or higher representing high risk [46].

In this study, NWO participants had larger WC measurements than lean young women. Our results demonstrated strong relationships between both total body FM and abdominal fat and cardiometabolic abnormalities. Similarly, Shea et al. [40] reported that the prevalence of cardiometabolic abnormalities was higher among participants with higher BFP compared to those with lower BFP. Our data demonstrate a statistically significant relationship between FBG and both BFP and WC. In a study by Kang et al. [9], total, visceral, and subcutaneous fat mass were all significantly correlated with vascular inflammation, which was greater than the traditional cardiovascular disease risk factors of age, BMI, and FBG levels. These findings suggest that both BMI and FM should be assessed to adequately screen participants with increased cardiovascular risk. It seems that specific screening of NWO might be necessary to prevent or treat cardiometabolic disease.

There is an increased risk for dyslipidemia with NWO [47]. Previous data suggest that HDL concentrations are significantly lower in people with NWO compared to lean individuals [48,49]. In support of previous studies, we observed that NWO was associated with an increased risk of low HDL concentrations compared to lean women. In terms of TG and LDL, there were no significant differences between NWO and the lean, overweight, type 1, and type 2 obese groups; however, the NWO group had significantly lower TG and LDL compared to the type 3 obese group. In support of our results, the first study that assayed lipid profile differences in NWO and non-NWO participants did not observe a significant difference in TG and LDL concentrations [14]. Conversely, a cross-sectional study of 3213 women and 2912 men observed that TG and LDL concentrations were higher in NWO participants compared to lean participants, and TG was found to be higher in people with obesity [15]. Furthermore, in a study by Shea et al., lean-weight participants were divided into sex-specific BFP tertiles as follows: low (≤15.2% men, ≤29.7% women), medium (15.3–20.7% men, 29.8–34.9% women), and high (≥20.8% men, ≥35.0% women) BFP [40]. They reported that although serum TG and LDL were higher in both the medium- and high-BFP groups, there were no significant differences in HDL cholesterol concentrations. Lastly, participants with NWO presented higher mean TG and LDL concentrations than non-NWO participants in a population-based birth cohort study [48].

It has been well documented that an excessive amount of FM is associated with numerous comorbidities, including hypertension and insulin resistance. While we did not observe any significant differences between NWO and lean participants for SBP, DBP, or HR, participants with a BMI higher than 35 had higher SBP compared to lean participants. The second Nurses’ Health Study was a prospective cohort study that reported an increased BMI resulted in women having a 4.7 times higher incidence of hypertension than women with a body mass index lower than 23.0 kg.m^−2^ [50]. In addition, based on the Framingham Offspring Study, obese women had a seven times higher risk of hypertension than lean women of the same age [51]. Our NWO results are in agreement with previous studies that demonstrated that there were no significant differences in BP between NWO, lean, and obese young participants [40,48]. However, other studies have reported differences [9,15]. The age of the participants is a potential explanation for this discrepancy, as participants used in previous studies had a mean age that was almost double the participants’ mean age in our study (27.2 years) [9,15]. Additional studies are needed to investigate BP differences between participants who are NWO, lean, overweight, and obese.

No significant differences were observed for FBG or insulin concentrations in NWO women compared to normal, overweight, and all three obese groups. It is commonly accepted that excessive body fat is associated with hyperglycemia and hyperinsulinemia [52]. However, these metabolic changes in NWO may be subclinical because there were no significant differences in FBG and insulin between young NWO and lean individuals [53]. In contrast, some studies revealed that NWO women showed higher FBG, fasting insulin, and insulin resistance levels compared to lean subjects [41,54]. For example, Romero-Corral et al. reported that NWO men and women have higher insulin resistance and beta-cell responses and lower insulin sensitivity compared to lean participants [41]. However, our data do not support these findings. Several factors may explain this difference. First, the population that was screened in the Romero-Corral et al. study was a mixture of women and men, whereas our participants were only women. Second, in the Romero-Corral et al. study, the definition of NWO was different from our study, as they classified NWO as a BFP higher than 33.3. Lastly, the mean age of our participants was approximately 10 years less than in the previous study. It is plausible that these factors explain the differences in the obtained results.

In a previous report [15], it was suggested that proinflammatory cytokines could be predictors of obesity risk, cardiovascular disease, and metabolic syndrome in NWO women. An earlier investigation showed that plasma proinflammatory cytokine concentrations were elevated in NWO, overweight, and obese women compared to lean women, and these concentrations were correlated with BFP [11]. Previous reports have observed NWO participants to have a higher subclinical atherosclerosis incidence than normal-weight lean individuals [55]. Hs-CRP is mainly used as a marker of subclinical chronic vascular inflammation and has a predictive value for future cardiovascular events [56]. In this study, no significant differences were found between NWO and lean women for hs-CRP concentrations, arguing against a relationship between NWO, inflammation, and cardiovascular disease. Our results indicated that, in young women, the risk of inflammatory-related disease is higher in women with type 2 obesity and above. Marques et al. assessed the levels of inflammation of NWO participants compared to lean, overweight, and obese participants. They reported no differences in hs-CRP concentrations between NWO and lean women, but they were significantly higher in overweight and obese participants [15]. In a cross-sectional study, NWO participants displayed hs-CRP concentrations 1.6 times higher compared to lean participants. However, this was not statistically significant [9]. In contrast, Tarantino et al. reported that the risk of having atherosclerotic plaque was higher in NWO compared to lean participants and that this was due to higher hs-CRP concentrations in NWO participants compared to lean participants [57]. When they divided the whole population by gender-specific BFP tertiles and defined NWO participants as those with a lean BMI and the highest tertile of BFP, the hs-CRP concentrations were significantly higher in NWO women compared to lean women.

Our data also indicate that NWO women present no significant impairment of liver enzymes. To date, research evaluating liver enzymes in populations with NWO is limited. It has previously been reported that no differences were found between NWO and lean women for concentrations of ALT, AST, and gamma-glutamyl transferase (GGT) [15]. Future studies may focus on the assessment of liver enzymes by echography among NWO women to better assess this point.

Our results showed that the risks associated with the metabolically abnormal phenotype were significantly higher among NWO individuals compared to lean participants. Although five of six cardiometabolic abnormality items (all phenotypes except HDL) were not significantly different between the NWO and lean categories, cardiometabolic dysregulation was significantly higher in NWO young women. Furthermore, this abnormality was elevated in type 2 and 3 obese participants compared to NWO participants. It has been reported that women with NWO have higher levels of cardiovascular risk factors than lean women [15]. The analysis from the NHANES III cohort reported that NWO is associated with a high prevalence of cardiometabolic dysregulation, metabolic syndrome, and cardiovascular risk factors [35]. In women, NWO is independently associated with an increased risk of cardiovascular mortality [41]. In addition, it was reported that NWO was an independent risk factor for the presence of soft plaques, even after further adjustment for multiple factors associated with atherosclerosis [55]. Therefore, it seems that NWO individuals carry a higher incidence of subclinical atherosclerosis compared with lean individuals, regardless of other clinical risk factors for atherosclerosis [57]. Moreover, according to our findings, cardiometabolic abnormalities were statistically the same between women with NWO and overweight and type 1 obesity. Although these results showed similar cardiometabolic abnormalities among women with NWO, overweight, and type 1 obesity, it should be mentioned that our study was the first study that compared metabolic abnormalities in six groups based on their respective BMI scores and BFP. More studies are needed to identify differences in cardiometabolic abnormalities between NWO, lean, overweight, and obese (types 1–3) participants.

This study has two major limitations that could be addressed by future research. First, the somewhat small population could be increased in future studies. Second, this study only focused on a female cohort. However, prior findings indicated that NWO is almost non-existent in men [15]. Future investigations focusing on understanding the relationship between NWO and cardiometabolic abnormalities warrant the evaluation of larger cohorts.

## 5. Conclusions

In summary, NWO women had higher body fat (FM and BFP) and lower lean mass (LBM%) compared with the lean group. Although no significant differences were found between the NWO and lean women for all cardiometabolic risk factors except HDL, compared to lean women, the increased BFP among NWO women was associated with increased cardiovascular abnormalities. NWO young women are at an increased risk of developing cardiometabolic disease, although the risks for type 2 and 3 obese individuals are still greater. The aforementioned cardiometabolic abnormalities were strongly related to BFP and waist circumferences. Specific screening of BFP may be necessary for all BMI categories to identify, prevent, and treat cardiometabolic disease at an early age in this population.

## Figures and Tables

**Figure 1 biomedicines-11-00341-f001:**
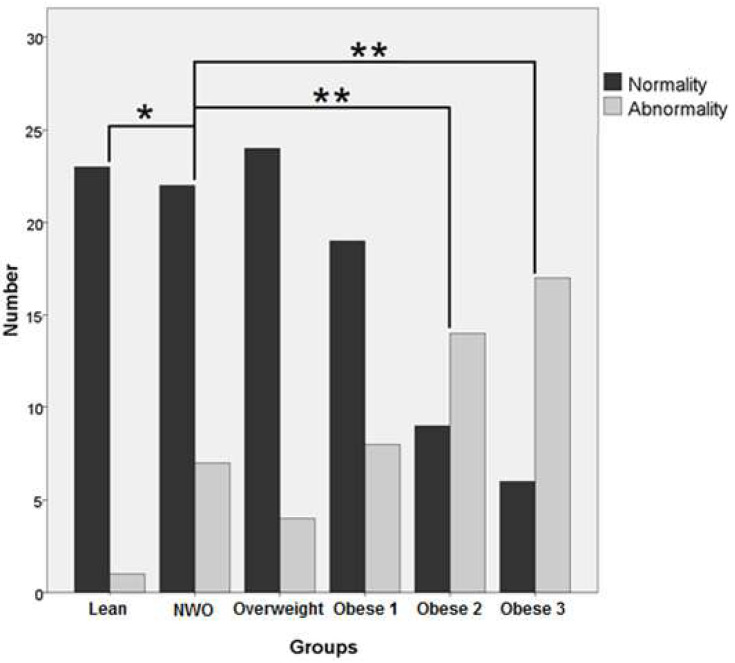
Comparison of the cardiometabolic abnormalities of NWO participants and those from other groups (percentage of abnormality: lean: 4.2%, NWO: 24.1%, overweight: 14.3%, obese type 1: 29.6%, obese type 2: 60%, and obese type 3: 73.9%). * *p* < 0.05 compared to NWO group; ** *p* < 0.001 compared to NWO group.

**Table 1 biomedicines-11-00341-t001:** Cardiometabolic biomarkers in the diagnosis of the metabolically abnormal phenotype.

Cardiometabolic Abnormalities Considered:
1. Elevated BP: systolic/diastolic BP ≥ 130/85 mm Hg or antihypertensive medication use;
2. Elevated TG: fasting TG ≥ 150 mg/dL;
3. Decreased HDL level: HDL level < 50 mg/dL in women or lipid-lowering medication use;
4. Elevated glucose level: FBG ≥ 100 mg/dL or antidiabetic medication use;
5. Insulin resistance: HOMA-IR > 5.13 (i.e., the 90th percentile);
6. Systemic inflammation: hs-CRP level > 0.1 mg/L (i.e., the 90th percentile).

Abbreviations: BP, blood pressure; TG, triacylglycerol; HDL, high-density lipoprotein cholesterol; FBG, fasting blood glucose; HOMA-IR, homeostatic model for insulin resistance; hs-CRP, high-sensitivity C-reactive protein.

**Table 2 biomedicines-11-00341-t002:** Anthropometric, body composition, and blood pressure characteristics in lean, normal-weight obese, overweight, and obese women.

Value	Lean	NWO	Overweight	Type 1 Obese	Type 2 Obese	Type 3 Obese	PV
**Age**	25.33 ± 5.37	27.21 ± 4.60	27.75 ± 5.30	28.63 ± 4.04	26.61 ± 4.08	28.30 ± 4.17	0.14
**Weight**	53.48 ± 5.70 ^**f, g, h, i**^	58.69 ± 4.58 ^**a, b, c, d, e**^	69.92 ± 4.48 ^**j, k, l**^	83.04 ± 6.23 ^**m, n**^	96.31 ± 6.30 ^**o**^	107.23 ± 8.77	<0.001
**Height**	160.42 ± 5.93	158.69 ± 4.58	159.39 ± 4.32	158.67 ± 4.67	160.84 ± 5.95	157.26 ± 5.30	0.18
**BMI**	20.75 ± 1.51 ^**f, g, h, i**^	23.29 ± 1.20 ^**a, b, c, d, e**^	27.50 ± 1.28 ^**j, k, l**^	32.93 ± 1.32 ^**m, n**^	37.85 ± 3.07 ^**o**^	43.35 ± 3.08	<0.001
**FM**	13.36 ± 1.97 ^**f, g, h, i**^	20.72 ± 2.13 ^**a, b, c, d, e**^	28.15 ± 2.68 ^**j, k, l**^	37.48 ± 3.25 ^**m, n**^	46.46 ± 4.01 ^**o**^	55.23 ± 5.41	<0.001
**BFP**	24.90 ± 1.81 ^**f, g, h, i**^	35.32 ± 2.45 ^**a, b, c, d, e**^	40.26 ± 2.79 ^**j, k, l**^	45.16 ± 2.76 ^**m, n**^	48.93 ± 2.31 ^**o**^	51.48 ± 2.07	<0.001
**LBM**	21.75 ± 2.39 ^**g, h, i**^	20.42 ± 2.05 ^**b, c, d, e**^	23.02 ± 2.36 ^**k, l**^	25.07 ± 2.65 ^**m, n**^	28.27 ± 5.68	29.04 ± 2.68	<0.001
**LBM%**	40.66 ± 0.98 ^**f, g, h, i**^	34.75 ± 1.63 ^**a, c, d, e**^	32.96 ± 3.28 ^**j, k, l**^	30.16 ± 1.75 ^**n**^	29.27 ± 4.85	27.09 ± 1.28	<0.001
**TBW**	29.43 ± 2.94 ^**g, h, i**^	27.76 ± 2.48 ^**b, c, d, e**^	30.53 ± 2.42 ^**j, k, l**^	33.35 ± 3.23 ^**m, n**^	36.66 ± 4.27	38.27 ± 3.21	<0.001
**FM/LBM**	0.61 ± 0.05 ^**f, g, h, i**^	1.02 ± 0.11 ^**a, b, c, d, e**^	1.23 ± 0.18 ^**j, k, l**^	1.50 ± 0.17 ^**m, n**^	1.69 ± 0.26 ^**o**^	1.90 ± 0.15	<0.001
**WHR**	0.74 ± 0.05 ^**f, g, h, i**^	0.77 ± 0.05 ^**c, d, e**^	0.80 ± 0.03 ^**j, k, l**^	0.85 ± 0.06 ^**n**^	0.89 ± 0.03	0.91 ± 0.04	<0.001
**WC**	67.52 ± 4.61 ^**f, g, h, i**^	74.25 ± 4.53 ^**a, b, c, d, e**^	83.76 ± 3.48 ^**j, k, l**^	96.00 ± 6.23 ^**m, n**^	106.19 ± 3.32 ^**o**^	112.30 ± 6.47	<0.001
**HC**	89.81 ± 3.77 ^**f, g, h, i**^	95.72 ± 4.48 ^**a, b, c, d, e**^	103.42 ± 4.54 ^**j, k, l**^	111.72 ± 5.15 ^**m, n**^	117.67 ± 4.19 ^**o**^	122.82 ± 5.76	<0.001
**HR**	98.33 ± 13.86	90.34 ± 13.42	91.71 ± 11.43	89.51 ± 13.30	94.04 ± 9.12	88.21 ± 14.41	0.08
**SBP**	111.56 ± 10.92 ^**h, i**^	115.70 ± 11.27	119.09 ± 16.14	117.74 ± 15.06	125.84 ± 16.53	124.15 ± 14.80	0.007

Lean, overweight, and obese women. Abbreviations: BMI, body mass index; FM, fat mass; BFP, body fat percentage; LBM, lean body mass; LBM%, LBM percentage; TBW, total body water; WHR, waist–hip ratio; WC, waist circumference; HC, hip circumference; HR, heart rate; SBP, systolic blood pressure; DBP, diastolic blood pressure. ^a^ *p* < 0.05 between NWO and lean; ^b^ *p* < 0.05 between NWO and overweight; ^c^ *p* < 0.05 between NWO and type 1 obese; ^d^ *p* < 0.05 between NWO and type 2 obese; ^e^ *p* < 0.05 between NWO and type 3 obese; ^f^ *p* < 0.05 between lean and overweight; ^g^ *p* < 0.05 between lean and type 1 obese. ^h^ *p* < 0.05 between lean and type 2 obese; ^i^ *p* < 0.05 between lean and type 3 obese; ^j^ *p* < 0.05 between overweight and type 1 obese; ^k^ *p* < 0.05 between overweight and type 2 obese; ^l^ *p* < 0.05 between overweight and type 3 obese; ^m^ *p* < 0.05 between type 1 obese and type 2 obese; ^n^ *p* < 0.05 between type 1 obese and type 3 obese; ^o^ *p* < 0.05 between type 2 obese and type 3 obese.

**Table 3 biomedicines-11-00341-t003:** Cardiometabolic risk factors in lean, normal-weight obese, overweight, and obese women.

Value	Lean	NWO	Overweight	Type 1 Obese	Type 2 Obese	Type 3 Obese	PV
**FBG**	84.87 ± 8.91 ^**i**^	89.27 ± 9.23	87.46 ± 10.35	90.55 ± 11.35	90.73 ± 9.56	94.56 ± 9.50	0.026
**Cholesterol**	167.16 ± 16.19 ^**i**^	179.89 ± 29.15	171.75 ± 27.18 ^**l**^	183.81 ± 34.95	181.73 ± 27.66	200.21 ± 36.87	0.004
**TG**	97.79 ± 23.53 ^**i**^	102.79 ± 64.52 ^**e**^	107.96 ± 43.78 ^**l**^	116.74 ± 41.89	133.43 ± 53.23	149.56 ± 49.71	0.002
**HDL**	55.91 ± 5.72 ^**f, g, h, i**^	49.10 ± 7.59 ^**a, d, e**^	47.78 ± 10.03 ^**l**^	44.55 ± 8.70	43.30 ± 8.90	37.91 ± 6.67	<0.001
**LDL**	93.83 ± 12.67 ^**h, i**^	107.20 ± 28.71 ^**e**^	102.57 ± 22.59 ^**l**^	114.14 ± 32.72	119.52 ± 25.27	134.78 ± 36.34	<0.001
**LDL/HDL**	1.69 ± 0.30 ^**g, h, i**^	2.28 ± 0.94 ^**e**^	2.29 ± 0.70 ^**l**^	2.63 ± 0.85 ^**n**^	2.95 ± 0.88 ^**o**^	3.78 ± 1.33	<0.001
**Cholesterol/HDL**	3.01 ± 0.34 ^**g, h, i**^	3.82 ± 1.30 ^**e**^	3.80 ± 0.91 ^**l**^	4.23 ± 0.95 ^**n**^	4.45 ± 1.04 ^**o**^	5.58 ± 1.54	<0.001
**VLDL**	19.55 ± 4.70 ^**i**^	20.55 ± 12.90 ^**e**^	21.59 ± 8.75 ^**l**^	23.34 ± 8.37	26.68 ± 10.64	29.91 ± 9.94	0.002
**AST**	21.50 ± 7.00	26.55 ± 24.10	23.21 ± 8.52	24.88 ± 13.73	22.60 ± 10.05	24.04 ± 9.76	0.823
**ALT**	18.79 ± 6.08	21.06 ± 13.94	23.21 ± 10.02	27.85 ± 16.27	22.26 ± 9.69	21.95 ± 7.74	0.113
**ALP**	175.50 ± 48.85	185.58 ± 37.39	168.21 ± 59.05	203.18 ± 61.01	194.60 ± 65.09	193.13 ± 59.99	0.212
**Insulin**	8.49 ± 3.48 ^**i**^	10.69 ± 5.96	11.54 ± 7.54	10.55 ± 5.02	11.01 ± 4.81	14.73 ± 9.98	0.045
**HOMA-IR**	1.07 ± 0.43 ^**i**^	1.36 ± 0.74	1.46 ± 0.93	1.35 ± 0.62	1.41 ± 0.59	1.89 ± 1.26	0.032
**HOMA-S percentage**	104.10 ± 33.89	95.85 ± 53.09	94.73 ± 55.06	89.13 ± 40.23	83.51 ± 35.43	78.33 ± 48.55	0.43
**HOMA-B percentage**	116.23 ± 37.18	121.72 ± 56.15	132.47 ± 63.51	119.90 ± 48.56	122.02 ± 50.76	130.15 ± 55.94	0.877
**hs-CRP**	0.93 ± 1.11 ^**h, i**^	2.08 ± 1.10 ^**d, e**^	1.81 ± 1.36 ^**k, l**^	3.22 ± 2.90 ^**m, n**^	7.73 ± 7.09	9.12 ± 5.59	<0.001

Abbreviations: FBG, fasting blood glucose; TG, triglyceride; HDL, high-density lipoprotein; LDL, low-density lipoprotein; VLDL, very low density lipoprotein; AST, aspartate aminotransferase; ALT, alanine aminotransferase; ALP, alkaline phosphatase; HOMA-IR, homeostatic model assessment of insulin resistance; HOMA-B, HOMA-pancreatic beta-cell function; HOMA-S, HOMA insulin sensitivity; hs-CRP, high-sensitivity C-reactive protein. ^a^ *p* < 0.05 between NWO and lean; ^b^ *p* < 0.05 between NWO and overweight; ^c^ *p* < 0.05 between NWO and type 1 obese; ^d^ *p* < 0.05 between NWO and type 2 obese; ^e^ *p* < 0.05 between NWO and type 3 obese; ^f^ *p* < 0.05 between lean and overweight; ^g^ *p* < 0.05 between lean and type 1 obese; ^h^ *p* < 0.05 between lean and type 2 obese; ^i^ *p* < 0.05 between lean and type 3 obese; ^j^ *p* < 0.05 between overweight and type 1 obese; ^k^ *p* < 0.05 between overweight and type 2 obese; ^l^ *p* < 0.05 between overweight and type 3 obese; ^m^ *p* < 0.05 between type 1 obese and type 2 obese; ^n^ *p* < 0.05 between type 1 obese and type 3 obese; ^o^ *p* < 0.05 between type 2 obese and type 3 obese.

**Table 4 biomedicines-11-00341-t004:** Partial correlations between BFP and WC and cardiometabolic abnormalities.

Variable	BFP		WC	
	R	*p*	r	*p*
SBP	0.279	0.000	0.286	<0.001
DBP	0.114	NS	0.105	NS
FBG	0.240	0.003	0.240	0.003
Cholesterol	0.247	0.002	0.253	0.002
TG	0.299	<0.001	0.313	<0.001
HDL	−0.599	<0.001	−0.511	<0.001
LDL	0.376	<0.001	0.359	<0.001
HOMA-IR	0.201	0.012	0.188	0.02
hs-CRP	0.518	<0.001	0.550	<0.001

Abbreviations: FBG, fasting blood glucose; TG, triglyceride; HDL, high-density lipoprotein; LDL, low-density lipoprotein; HOMA-IR, homeostatic model assessment of insulin resistance; hs-CRP, high-sensitivity C-reactive protein; NS, non-significant.

## Data Availability

The datasets used and/or analyzed during the current study are available from the corresponding author upon reasonable request.

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
