# Peer review of "Are Women with Normal-Weight Obesity at Higher Risk for Cardiometabolic Disorders?"

_biomedicines, 2023, doi:10.3390/biomedicines11020341_

Round 1

Reviewer 1 Report

There are many ways to assess body composition. How does your chosen research method (the direct multi-frequency bioelectrical impedance method) correlate with other methods?

Author Response

There are many ways to assess body composition. How does your chosen research method (the direct multi-frequency bioelectrical impedance method) correlate with other methods?

Authors: Previous studies showed a strong correlation between DXA and doubly labeled water (DLW) results for body composition and those obtained by BIA (1; 2; 3).

Reviewer 2 Report

In this manuscript, Ashtary-Larky et al investigated the anthropometric and cardio metabolic profiles of young women with NWO and also assessed the relationship between BFP and cardio metabolic abnormalities in young women with NWO, in comparison with lean, overweight and obese women. To this aim the authors performed a cross-sectional study that assessed cardio metabolic abnormalities of women with NWO compared to lean, overweight and obese women. The authors also examined various anthropometric variables, cardio metabolic abnormality markers (blood pressure, triglyceride, high density lipoprotein (HDL), fasting glucose, insulin resistance and high sensitivity C-reactive protein (hs-CRP)) and liver parameters were also examined. In summary they found that the NWO group had a significantly higher incidence of cardio metabolic abnormalities compared to lean women; whereas type 2 and 3 obese individuals displayed higher incidences compared to the NWO group. The researches concluded that women with NWO had a significantly higher incidence of cardio metabolic abnormalities when compared to lean subjects and that these abnormalities strongly relate to BFP and waist circumferences. This is a well written manuscript in which overall the results support the authors conclusions. However, there are some issues that require clarification.

Main issues:

1. The authors state that “Anthropometrics, body composition, and BP of all groups are shown in Table 2.” However, Table 2 does not show these results but rather it is a duplication of Table 3, titled “Cardiometabolic risk factors in lean, normal weight obese, overweight and obese women.” The authors need to provide the actual Table 2 results.

2. The size of the study groups is rather small as the authors acknowledge. I think that this is a major limitation of the study; however, the authors state that this is a “minor limitation”. Could the authors elaborate on this point of view?

3. In Figure 1, the overweight group displays higher normality/abnormality ratio than NWO despite the fact that overweight subjects are a step up in fat content.

Author Response

  1. The authors state that “Anthropometrics, body composition, and BP of all groups are shown in Table 2.” However, Table 2 does not show these results but rather it is a duplication of Table 3, titled “Cardiometabolic risk factors in lean, normal weight obese, overweight and obese women.” The authors need to provide the actual Table 2 results.

Authors: It was a typo. Table 2 was corrected.

  1. The size of the study groups is rather small as the authors acknowledge. I think that this is a major limitation of the study; however, the authors state that this is a “minor limitation”. Could the authors elaborate on this point of view?

Authors: Yes, you are right. It is a major limitation of our study. Corrections were made.

  1. In Figure 1, the overweight group displays higher normality/abnormality ratio than NWO despite the fact that overweight subjects are a step up in fat content.

Authors: You are right. The overweight group displays a higher normality/abnormality ratio than NWO. However, there is not any statistical significance between these two groups. Therefore, we cannot conclude that NOW has higher cardiometabolic abnormality compared with overweight.

  1. English language and style are fine/minor spell check require

Authors: Thank you for this point. Our manuscript was reviewed by three native speakers and we think the quality has vastly improved. Thank you once again.

  1. Is the research design appropriate? Can be improved

Authors: Corrections were made.

  1. Are the results clearly presented? Must be improved

Authors: We would like to kindly mention that we tried to improve the quality of the presented results. Some are presented in the table and some are within the main text. If you still think we should add more results, we would be thankful if you point them out in detail. Thank you.

  1. Are the conclusions supported by the results? Can be improved

Authors: Corrections were made in the conclusion.

Reviewer 3 Report

1. 2.2 The definition of type 3 obese women is wrong, it should be corrected.

2. Tables 2 and 3 are exactly the same.

3. The authors should give a more logical explanation for the following results.

"No significant differences were observed for FBG or insulin concentrations, in NWO women compared to normal, overweight, and all obese groups."

Author Response

  1. 2 The definition of type 3 obese women is wrong; it should be corrected.

Authors: Done.

  1. Tables 2 and 3 are exactly the same.

Authors: It was a typo. Corrections were made.

  1. The authors should give a more logical explanation for the following results.

"No significant differences were observed for FBG or insulin concentrations, in NWO women compared to normal, overweight, and all obese groups."

Authors: More explanations were added.

  1. English language and style are fine/minor spell check required

Authors: Our manuscript was reviewed by three native speakers and we think the quality has vastly improved. Thank you once again.

Reviewer 4 Report

Congratulations to the authors, a very interesting article and useful in clinical practice.

Comments:

- Citations and bibliography do not respect the recommendations of the journal

- The term normal weight in Asian people – BMI < 25 kg/m2 or BMI<23 kg/m2?

- Line 353: Regarding the assessment of liver function, this is not complete. Besides TGP, TGO and ALP levels, imaging techniques should also be used - ultrasound, FibroScan, etc. The new criteria of metabolic dysfunction -associated fatty liver disease: MAFLD in people with lean/normal weight - BMI < 23kg/m2 in Asians

- Eslam M et al - A new definition for metabolic dysfunction-associated fatty liver disease: An international expert consensus statement, 2020

- Chen F et al Lean NAFLD: a distinct entity shaped by differential metabolic adaptation, 2020

Author Response

- Citations and bibliography do not respect the recommendations of the journal

Authors: Corrections were made throughout the text.

- The term normal weight in Asian people – BMI < 25 kg/m2 or BMI<23 kg/m2?

Authors: We used the international classification for BMI since the definition of NOW is based on this classification (18.5≤BMI<25 and BFP >30%).

- Line 353: Regarding the assessment of liver function, this is not complete. Besides TGP, TGO and ALP levels, imaging techniques should also be used - ultrasound, FibroScan, etc. The new criteria of metabolic dysfunction -associated fatty liver disease: MAFLD in people with lean/normal weight - BMI < 23kg/m2 in Asians

Authors: “Liver enzymes” was replaced by live function through the text. As mentioned above, since the definition of NOW is based on BMI international classification (18.5≤BMI<25 and BFP >30%), therefore, we used the international classification for BMI.

- Eslam M et al - A new definition for metabolic dysfunction-associated fatty liver disease: An international expert consensus statement, 2020

- Chen F et al Lean NAFLD: a distinct entity shaped by differential metabolic adaptation, 2020

- Are the results clearly presented? Can be improved

Authors: We would like to kindly mention that we tried to improve the quality of the presented results. Some are presented in the table and some are within the main text. If you still think we should add more results, we would be thankful if you point them out in detail. Thank you.

  1. Lahav Y, Goldstein N, Gepner YJEjocn (2021) Comparison of body composition assessment across body mass index categories by two multifrequency bioelectrical impedance analysis devices and dual-energy X-ray absorptiometry in clinical settings. 75, 1275-1282.
  2. Siedler MR, Rodriguez C, Stratton MT et al. (2022) Assessing the Reliability and Cross-Sectional and Longitudinal Validity of 15 Bioelectrical Impedance Analysis Devices. 1-29.
  3. Beato GC, Ravelli MN, Crisp AH et al. (2019) Agreement between body composition assessed by bioelectrical impedance analysis and doubly labeled water in obese women submitted to bariatric surgery. 29, 183-189.

Round 2

Reviewer 2 Report

The authors have successfully addressed my main concerns. I have no further comments.

Reviewer 3 Report

I have no further comments